# Structure and Function of the Zinc Binding Protein ZrgA from *Vibrio cholerae*

**DOI:** 10.3390/ijms24010548

**Published:** 2022-12-29

**Authors:** Daniel W. Valencia, Ady B. Meléndez, Isaac A. Melendrez, Erik T. Yukl

**Affiliations:** Department of Chemistry and Biochemistry, New Mexico State University, Las Cruces, NM 88003, USA

**Keywords:** ABC transporters, solute-binding protein, zinc

## Abstract

ATP binding cassette (ABC) transporters are the primary means by which bacteria acquire trace elements from the environment. They rely on solute binding proteins (SBPs) to bind the relevant substrate and deliver it to the integral membrane permease for ATP-powered import into the cytoplasm. SBPs of cluster A-I are known to facilitate the transport of essential metals zinc, manganese, and iron, and many have been characterized to date. A group of ABC transporter operons dubbed zinc-regulated genes (*zrg*) have recently been shown to transport zinc with putative SBPs (*zrgA*) bearing no homology to the classical cluster A-I family, and a recent crystal structure of a representative protein from *Pseudomonas aeruginosa* shows no structural similarity to classical SBPs. Thus, the ZrgA proteins appear to represent a newly discovered family of zinc SBPs widespread among Gram-negative bacteria, including human pathogens. Here, we have determined the crystal structure of ZrgA from *Vibrio cholerae* and characterized its zinc binding in vitro and function in vivo. We also assessed the role of a histidine-rich sequence that appears to be a hallmark of ZrgA proteins that is particularly long in *V. cholerae* ZrgA. The results show that the *zrgA* gene is critical to the function of the operon, consistent with a function as an SBP in this system. Further, the His-rich region is not essential to the function of ZrgA, but it does provide additional zinc binding sites in vitro. The structure and zinc binding data for ZrgA reveal interesting differences between it and its homologue from *P. aeruginosa*, illustrating diversity within this little-studied protein family.

## 1. Introduction

The ATP binding cassette (ABC) transporters facilitate the transport of a tremendous variety of substrates across the cell membrane in all kingdoms of life [1]. Transport through an integral membrane permease is powered by ATP hydrolysis at a cytoplasmic ATPase. In addition to these proteins, prokaryotic ABC transporters tasked with importing various substrates require an extracellular solute binding protein (SBP) component [2]. The SBP may be periplasmic, membrane tethered or fused to the permease [3], and it functions to bind the substrate with high affinity and specificity and deliver it to the permease for import into the cytoplasm [4].

Despite limited sequence homology and widely varying substrate specificity, SBP structures are highly conserved with the substrate binding site located between two related α/β domains. These proteins have been classified into 7 clusters (A-G) [5,6] according to the nature of the interdomain linker with further subdivisions based on substrate specificity. Cluster A-I SBPs directly bind zinc, manganese or iron with high affinity and are often essential for survival in metal-limited environments. In the case of pathogens, the human host is such an environment [7], making these ABC transporters attractive targets for the development of novel antibiotics.

Cluster A-I SBPs have been intensively studied, and many crystal structures are available. An unusual feature of zinc-specific SBPs of this class is the presence of a flexible loop near the high-affinity binding site that differs in length and composition but is generally unstructured. In Gram-negative organisms, this loop is rich in His residues and can provide additional zinc binding sites [8,9,10,11,12,13,14], although its function remains enigmatic.

ABC transporter systems with non-canonical SBP genes have been implicated in zinc import in *Vibrio cholerae* [15] and *Pseudomonas aeruginosa* [10] in addition to the classical ZnuABC system where ZnuA is the cluster A-I SBP. In *V. cholerae*, such a transporter operon was annotated as zinc-regulated genes *zrgABCDE* and shown to be critical along with *znuABC* for optimal growth in zinc-limited conditions and virulence in a mouse model. The *zrgA* gene encodes a predicted periplasmic protein with no sequence similarity to known SBPs. However, it is conserved in hundreds of organisms in close proximity to ABC transporter ATPase and permease genes [16], *zrgC* and *zrgB*, respectively. These observations suggest a function for ZrgA as a novel zinc SBP, although its independent function in vivo has not been evaluated to our knowledge. A crystal structure for the ZrgA homologue in *P. aeruginosa*, annotated PA4063, was recently solved and exhibits no structural similarity to cluster A-I SBPs [17]. Two zinc binding sites were identified with low micromolar binding affinity, only one of which is conserved in *V. cholerae* ZrgA (site b, see Figure 1) or amongst other ZrgA homologues (Appendix A). Interestingly, one feature in common between ZrgA proteins and cluster A-I SBPs is the presence of a flexible, His-rich sequence of varied length and composition. This loop and the His-rich N-terminal region are disordered and not observed in the crystal structure of PA4063. Similar sequences are present in *V. cholerae* ZrgA and the central loop is especially rich in potential zinc ligands His, Asp, and Glu (Figure 1).

In this work, we set out to determine the role of *zrgA* in zinc import through the *zrgABCDE* system in *V. cholerae*. To that end, we generated a scarless *ΔzrgA* strain in a *ΔznuA* background and evaluated its growth and survival in zinc-deficient media. We also investigated a mutant lacking only the flexible loop (*zrgA Δ*124-184) to assess the function of this feature. The WT and deletion mutants of the ZrgA protein were expressed and purified, and their zinc binding properties were characterized. The crystal structure of ZrgA Δ124-184 was solved and shows a conserved structure with PA4063 but exhibits interesting differences in zinc binding. Overall, this work provides compelling evidence for ZrgA as a new family of zinc SBPs and illustrates diversity in zinc binding properties even between the only two characterized members.

## 2. Results

### 2.1. ZrgA Is Essential to Zinc Import through zrgABCDE and Does Not Require the Flexible Loop

Several mutants of *V. cholerae* zinc ABC transporters were generated and evaluated for their ability to grow and survive in zinc-deficient media (Figure 2). These include a deletion of the gene encoding the cluster A-I SBP ZnuA (*ΔznuA*), a double knockout of both putative zinc SBPs (*ΔznuA/ΔzrgA*), and a mutant expressing ZrgA lacking the His-rich loop in the ZnuA-null background (*ΔznuA/zrgA Δ124-184*). All three mutants exhibited a modest growth lag in zinc-replete media relative to WT, which was exacerbated in zinc-deficient media. However, all strains reached comparable densities after 24 h of growth in either condition with the exception of the double knockout *ΔznuA/ΔzrgA* under zinc-deficient conditions (Figure 2A). This is confirmed by colony-forming units showing an approximately 100-fold decrease in survival for this mutant in zinc-deficient conditions (Figure 2C). These results are consistent with those previously observed indicating that both *znu* and *zrg* systems function in zinc import and that *znu* is the primary zinc importer in media [15]. They further show that *zrgA* is essential to the function of the *zrgABCDE* operon and that the His-rich loop is a non-essential feature of the protein. 

### 2.2. Expression, Purification and Zinc Binding of WT and Loop Deletion Mutants of ZrgA

The full-length *zrgA* gene from *V. cholerae* was expressed in *E. coli* and the protein was purified from the periplasm using ion exchange and size exclusion chromatography. We also expressed and purified several deletion mutants lacking residues of the His-rich loop (Δ124-184 ZrgA, Δ124-183 ZrgA, and Δ124-180 ZrgA, see Figure 1) using the same protocol. Purified proteins were analyzed for zinc content using inductively couple plasma optical emission spectroscopy (ICP-OES). The results from three independent preparations show that WT ZrgA purifies with 0.44 ± 0.10 equivalents of zinc whereas zinc was undetectable in any mutant protein as purified. Zinc was readily removed from the WT protein by dialysis in acidic conditions in the presence of EDTA. 

To determine the zinc binding affinity and stoichiometry, a competitive binding assay with the fluorophore MagFura-2 (MF2) was employed (Figure 3, Table 1). The data for WT and Δ124-180 ZrgA best fit a model with 4 high-affinity binding sites, while only 3 sites were required to fit the data for the other two, longer deletion mutants. Presumably, Glu 181 and/or His 182 are involved in zinc binding. Measured zinc binding affinities range from 0.03–287 nM. As this is a competition assay, Kd values less than approximately 100-fold that of MF2 (Kd = 36 nM [20]) are difficult to estimate while binding sites with Kd over 100-fold higher are not observed. To determine the total zinc binding capacity of WT and Δ124-184 ZrgA, different concentrations of zinc were added to each apo protein followed by desalting and ICP-OES to determine bound zinc content (Appendix A). The results indicated that WT ZrgA can bind up to 10 zinc ions while Δ124-184 ZrgA saturates around 5 or 6 ions. The linearity of the pre-saturation portion of the graphs relating zinc concentration to bound equivalents suggests that the binding affinity at these additional sites is in the low to the mid-micromolar range, considering the starting protein concentrations of 100 μM and the fact that they are not observed by the MF2 assay.

### 2.3. Crystal Structure of V. Cholerae ZrgA Δ124-184

Despite multiple attempts, we were unable to obtain diffraction quality crystals of WT ZrgA. We were able to crystallize Δ124-184 ZrgA but only in the absence of zinc. The apo protein structure (Figure 4A) was solved in the *P*2_1_ space group to a resolution of 2.00 Å (Table 2) by molecular replacement using the *P. aeruginosa* homologue PA4063 (PDB ID: 7AHW) as a search model. Six protein molecules occupy the asymmetric unit, although the protein is purified exclusively as a monomer as determined by size exclusion chromatography (Appendix A). The N-terminal residues from Asp 22, the first after the predicted signal sequence cleavage site [18,19] (Figure 1), to Gly 39 could not be modeled into any chain due to a lack of electron density. The same is true of residues 113–123 comprising the first part of the flexible loop, except for chain D where only residues 116–120 are missing. These regions are also absent in the apo PA4063 structure, indicating a high degree of flexibility and disorder.

A disulfide bond is present between Cys 100 and Cys 195 in ZrgA that is conserved in PA4063 and other homologues (Appendix A). The tertiary structures of ZrgA and PA4063 are very similar with overall rmsd ~ 1.2 Å^2^ between backbone atoms (Appendix A). The beta-sandwich structures are virtually superimposable, although the long alpha helix is angled away from the central beta sheet by a few degrees in ZrgA relative to PA4063. Overall, the ferredoxin-like fold of ZrgA family proteins appears to be highly conserved.

Since zinc-bound ZrgA would not crystallize, zinc was soaked into crystals of apo Δ124-184 ZrgA to acquire the holo protein structure at 2.41 Å resolution. We also acquired a dataset at the zinc absorption edge to generate an anomalous density map, which allowed us to model a total of 21 zinc ions into the asymmetric unit. These fall into 5 different binding sites (Appendix A). Sites 1 and 2 are observed in every chain with strong anomalous density and relatively low B-factors for the zinc ion (Figure 4B). Site 3 is occupied in all chains as well, although it is not always refined at full occupancy due to high B-factors and weak anomalous density. Site 3b and site 4 are only observed in some chains at sub-stoichiometric occupancy.

Site 1 (Figure 5A,D) is coordinated by the previously disordered His 35 and His 37 as well as Glu 66. The latter is modeled in a bidentate binding mode in most chains, although we feel that the resolution of the data precludes a definitive assignment of binding mode. Site 1 is conserved in PA4063 where a fourth His aligning with His 184 in ZrgA coordinates the metal. However, this residue is absent in the ZrgA deletion mutant and is not observed, although its position suggests it likely serves as a ligand in the WT ZrgA as it does in PA4063. A water molecule appears to occupy the fourth binding site in most chains. 

Site 2 (Figure 5B,E) is observed on the long alpha helix and is shared between two chains, one containing ligands Glu 83, His 86 and His 87, the other contributing Glu 79. Some variation between monodentate and bidentate binding is observed with the Glu residues in different chains. These residues are not conserved in other ZrgA proteins (Appendix A) including PA4063 nor is zinc observed to bind near this position in the PA4063 structure.

Site 3 (Figure 5C,F) is located at the opposite end of the protein from site 1 and is coordinated by residues His 49, His 194, and Glu 196. In chains A and E, His 118 from a neighboring chain serves as the fourth ligand. This interaction stabilizes part or the entire loop comprising residues 113–123, which could not be modeled into the apo structure. In other chains, the fourth position may be occupied by a labile water molecule. In chains C and D, a second zinc is observed nearby, bound by residues His 194 and Asp 50, which we label site 3b (Appendix A). Weak anomalous density and high B-factors of 3 and 3b sites contained within the same chain may indicate that these sites are not occupied simultaneously with His 194 acting as a bridging ligand. Thus, we set the occupancies of these zinc atoms as well as any others with unusually high B-factors to less than one and allowed them to refine. The results suggest that we observe the average of occupied site 3 or site 3b in some ZrgA molecules. Neither binding site is observed in PA4063. A final zinc binding site 4 (Appendix A) was identified by anomalous density. The only ligand observed to coordinate the metal is His 67 of chain B. Here, again, the occupancy of this metal is sub-stoichiometric (Appendix A).

Although the structures of apo and holo ZrgA are very similar with RMSD ~ 0.5 Å, they do reveal a few interesting conformational changes accompanying zinc binding (Figure 6), particularly near site 1. As was observed for PA4063, site 1 ligands His 35 and His 37 become ordered upon metal binding. Further, the loop containing Glu 66 moves approximately 4 Å to allow zinc coordination (Figure 6B). In order to accommodate this motion, the N-terminal side of the flexible loop (modeled residues 121–123) must also reorient a comparable distance. The side chains of site 2 ligands Glu 83 and His 86 reorient on zinc binding as well (Figure 6C) while only very modest side chain movement is observed at site 3 (Figure 6D). 

## 3. Discussion

Previously, knockouts of the entire *zrgABCDE* operon or the *zrgABC* genes were evaluated for growth under zinc limitation imposed by the chelator TPEN [15] to confirm a function in zinc import. However, the independent function of *zrgA* was not evaluated. Here, we show that ZrgA likely functions as the SBP for the ZrgBC permease/ATPase transporter system as it is required for normal growth and survival in zinc-deficient conditions in the *ΔznuA* background. The *zrgD* gene encodes a hypothetical, 101-amino acid protein with a single transmembrane helix while *zrgE* encodes a hypothetical periplasmic protein of 150 amino acids after signal sequence cleavage. Individual *zrgD* or *zrgE* knockouts had no effect on growth in the presence of zinc chelators, although the *zrgABCDE* knockout did exhibit a very slightly more severe growth deficient phenotype than *zrgABC*, suggesting these protein products may facilitate zinc transfer through the *zrgABC* system [15]. A Structural Genomics structure for the *P. aeruginosa* homologue of ZrgE (PA4066) with 40% sequence identity to *V. cholerae* ZrgE has been solved but provides no clue as to its function. In any case, our data demonstrate that ZrgE cannot substitute for ZrgA, establishing the latter as the most likely candidate for a periplasmic SBP to the ZrgBC core transporter system.

In vitro, WT ZrgA can bind up to 10 zinc ions (Appendix A) with four high-affinity (Kd < 1 μM) zinc binding sites (Table 1). The unstructured, His-rich loop accounts for four or five binding sites (Appendix A), most of which are of relatively low affinity. However, residues Glu 181 and His 182 of the loop likely participate in high-affinity binding as four high affinity binding sites are observed in the Δ124-180 ZrgA mutant while only three are observed in Δ124-183 or Δ124-184 ZrgA. In every ZrgA construct measured, the highest affinity site has a Kd of less than 1 nM. Affinities in the low to sub-nM range are not unusual among the cluster A-I zinc SBPs [8,9,10,11,13,14,21,22], allowing them to support growth and survival under extremely zinc-limited conditions as is observed here for ZrgA. In contrast, PA4063 is reported to bind only 2 zinc ions with comparable affinities of ~2 μM as determined by isothermal titration calorimetry (ITC) [17], several orders of magnitude weaker binding than is observed for ZrgA. The central His-rich loop is much shorter in PA4063 relative to ZrgA, which likely accounts for the decreased zinc binding capacity. However, the relatively weak binding affinity is puzzling for a SBP, leading to the hypothesis that this protein may act as a sensor, negatively regulating zinc import under conditions of excess zinc [17]. To our knowledge, there is no in vivo evidence to suggest either function as the PA4063-PA4066 operon remains uncharacterized. Nevertheless, it is intriguing that these two closely related proteins should exhibit such different metal binding properties and potentially even divergent functions.

The crystal structure of apo Δ124-184 ZrgA was nearly identical to PA4063, including the observation of unstructured regions at the N-terminus and His-rich loop. As for PA4063, zinc binding led to partial ordering of these regions, particularly His35 and His37, which bind zinc along with Glu66 at site 1 (site b in PA4063). This is the only shared zinc site between the two protein structures, although Δ124-184 ZrgA is missing the His184 residue predicted to complete the coordination sphere at site 1. It is also the only zinc binding site that is widely conserved among ZrgA homologues (Appendix A). These observations combined with the high zinc occupancy and strong anomalous density at most site 1 positions make it tempting to assign this as the highest affinity binding site. If correct, this would indicate that loss of the His184 ligand and nearby residues does not significantly decrease binding affinity, although it may increase the off-rate for zinc, resulting in its loss during purification. Similar properties were observed for the cluster A-I SBP AztC, where a mutant with two His ligands replaced by Ala still bound zinc with high-affinity but exhibited a much higher off-rate [23]. Further, the position of site 1 at the end of the ZrgA molecule places it at a potential protein binding interface promoting facile transfer to the putative permease ZrgB. Finally, the only significant structural changes accompanying zinc binding occur near site 1. It is likely that the permease must be able to differentiate apo and holo ZrgA structures in order to promote dissociation after metal transfer and prevent the formation of unproductive complexes with apo ZrgA. Further mutagenesis work will be required to determine the functions of different zinc binding sites in ZrgA.

Site 2 is observed only in ZrgA and is not conserved in ZrgA homologues. However, it is also highly occupied in the crystal structure, suggesting relatively high affinity. Sites 3 or 3b are also present in every chain, but probably not simultaneously as indicated by sub-stoichiometric occupancies at chains containing metals at both 3 and 3b sites. In any case, the binding data in solution agree well with the crystal structure of Δ124-184 ZrgA having 3 high-affinity binding sites. Site 4 is observed in only one chain bound to a single His67 ligand, suggesting a relatively low binding affinity not observed by MF2 assay. The function of these other binding sites along with the multiple binding sites ascribed to the His-rich loop also remain to be elucidated. 

ZrgA adopts a ferredoxin-like fold, albeit with different topology from the classic ferredoxin (fd), that bears no similarity to known SBPs. While zinc binding and regulatory functions are known for fd-like folds in the ZntA and cation diffusion facilitator (CDF) families [24,25], our current data suggest that zinc SBP can be added to the long list of functions that fd-like folds have evolved to fill. The membrane permease ZrgB also exhibits significant differences from known metal-transporting permeases. PsaC, the only structurally characterized zinc/manganese ABC transporter permease, has 9 transmembrane (TM) helices and virtually no periplasmic domain [26]. In contrast, ZrgB is predicted to have 4-TM helices and a large periplasmic domain between TM helices 1 and 2 similar to MacB [27], which transports the enterotoxin STII from the periplasm to the extracellular space [28]. This brings up the intriguing possibility that the ZrgB periplasmic domain may interact with ZrgA and/or other elements of the ZrgABCDE system. Thus, we expect that the structures and binding interfaces in the Zrg family transporters will provide unique insight into the mechanisms of bacterial zinc import distinct from what has been shown for other ABC transporters. This knowledge will be particularly important given the role of ZrgA in *V. cholerae* virulence [15] and its conservation in established pathogens such as *P. aeruginosa* and emerging pathogens such as those from the genus *Aeromonas* [16,29]. 

## 4. Materials and Methods

### 4.1. Multiple Sequence Alignments

The protein sequence of ZrgA from *V. cholerae* (UniProtKB accession number Q9KP27) was used to perform a BLAST search of the UniProtKB database [30]. A small group of protein sequences with E values below 10^−20^ were selected for multiple sequence alignment in Clustal Omega [31] and displayed in Jalview [32]. 

### 4.2. Strains and Media 

See Appendix A for the list of strains and primers used, respectively. *E. coli* was grown at 37 °C in Luria-Bertani (LB) and *V. cholerae* O395 at 30 °C in minimal medium. The minimal medium was made by combining the following components at the specified concentration and adjusting the pH to 7.4: 1X M9 medium, 1X trace elements solution, 0.4% *w*/*v* glucose, 1 mM MgSO_4_, 0.3 mM CaCl_2_ and 0, 10 or 50 µM ZnSO_4_. The 10X M9 medium stock was prepared by dissolving in Milli-Q water: 0.56 M Na_2_HPO_4_, 0.29 M KH_2_PO_4,_ 85.5 mM NaCl and 93.47 mM NH_4_Cl, adjusting pH to 7.5. For the 100X trace elements solution stock: 17.1 mM EDTA, 3.07 mM FeCl_3_·6H_2_O, 76.3 µM CuCl_2_·2H_2_0, 42 µM CoCl_2_·6H_2_O, 162 uM H_3_BO_3_, 6.84 µM MnCl_2_·6H_2_O were dissolved in water. Additionally, stocks solutions of 100 mM ZnSO_4,_ 1M MgSO_4_, 1M CaCl_2_ and 20% glucose were independently prepared. All were filter-sterilized into plastic containers to avoid trace zinc contamination from glassware.

### 4.3. Construction of V. cholerae Deletion Mutant Strains

Scarless deletions of *znuA* and *zrgA* genes as well as ZrgA residues 124–184 were generated through PCR amplification of 500–600 bp flanking sequences of the target genes/regions and assembling into the *NcoI* digested suicide vector pGP704sacB28 (Ap^r^) [33] using the Gibson cloning method [34] and confirmed with PCR and sequencing. Then, the plasmid was transformed into Sm10 λ pir cells for mating into *V. cholerae* O395 cells. Cells were streaked on LB agar plates with 100 μg/mL ampicillin and 50 μg/mL streptomycin to isolate *V. cholerae* transformants. Since pGP704sacB28 contains the counter-selectable sucrose sensitivity marker *sacB*, *V. cholerae* double recombinants were selected by streaking cells in 6% sucrose LB agar. Verification of double crossover recombinants after sucrose selection was performed by replica plating on LB agar containing 50 μg/mL streptomycin and LB agar containing 50 μg/mL streptomycin and 100 μg/mL ampicillin to confirm the loss of ampicillin resistance. Scarless deletions were confirmed through PCR and sequencing of genomic DNA around the deletion site.

### 4.4. Cell Growth Determination

Overnight cultures of *V. cholerae* WT and mutant cells from frozen stocks were plated on LB streptomycin plates. Starter cultures were grown from a single colony in minimal media containing 10 µM ZnSO_4._ The pellet collected from 500 µL of overnight cultures was washed twice, resuspended in zinc-depleted media, and inoculated into 96-well microplates to an OD_600_ of 0.02 in 200 µL of minimal media with no zinc added or 50 µM ZnSO_4_. Cell growth at 30 °C was monitored for 24 h with orbital shaking of 225 rpm in the BioTek EPOCH microplate reader.

### 4.5. Colony Forming Units

For CFU/mL quantification, cultures were diluted 10^4^-fold to 10^7^-fold and plated on LB agar (1.5% agar) with 50 µg/mL streptomycin in 150 mm × 15 mm Petri dishes with a glass spreader. Plates were incubated for 24 h at 30 °C. Colonies were quantified from pictures of plates using Fiji software with the cell counter plugin [35].

### 4.6. Expression and Purification of ZrgA and Mutant Variants

The intact *zrgA* gene was amplified from *V. cholerae* strain N16961 genomic DNA and cloned into the *ndeI* and *acc65I* restriction sites of the pCDF-Duet1 expression vector to generate the full-length, untagged protein. The ∆124-184, ∆124-183 and ∆124-180 *zrgA* mutants were generated by site-directed mutagenesis. Plasmids were transformed into BL21 *E. coli* cells and grown with shaking at 37 °C in LB media containing 50 μg/mL streptomycin to an OD_600_ of 0.8–1.0 and then induced with 1 mM IPTG and incubated with shaking at 20 °C for 20 h. The cells were collected by centrifuging at 4000× *g* for 30 min at 4 °C. The periplasmic fraction was obtained using an osmotic shock protocol adapted from Wang et al. [36], and ZrgA was purified using anion exchange chromatography and size exclusion chromatography as previously described [37]. Protein concentrations were determined using an extinction coefficient at 280 nm of 11,092 M^−1^cm^−1^ calculated as previously described [38] 

### 4.7. Generation of Apo Proteins and Metal Quantitation

Apo proteins were generated by dialysis at 4 °C against one change of 50 mM sodium acetate buffer pH 4.5, 50 mM ethylenediaminetetraacetate (EDTA), and 150 mM NaCl. This was followed by dialysis against two changes of 50 mM tris buffer pH 8.0, 150 mM NaCl, and 3.6 g/L Chelex resin (Bio-Rad). Zinc content of proteins as isolated as well as apoproteins was determined by inductively coupled plasma-optical emission spectroscopy (ICP-OES). Briefly, protein samples at a concentration of 50 μM were digested in concentrated HNO_3_ overnight at 70 °C and diluted to 3 mL with Milli-Q water prior to metal analysis. Buffer blanks were generated identically using an equal volume of the buffer relative to the protein solution. Metal content was quantified using a PerkinElmer 2100 DV ICP-OES, calibrated with a multielement standard (Alpha Aesar) at a wavelength of 213.857 nm for zinc.

### 4.8. Mag-Fura 2 Competition Assay

Zinc binding affinities were measured using a Mag-Fura 2 (MF2, Invitrogen, Carlsbad, CA, USA) competition assay derived from Golynskiy et al. [20] as previously described [37,39]. All fluorescence measurements were made using a Varian Cary Eclipse (Palo Alto, CA, USA) fluorescence spectrophotometer with entrance slit set to 10 nm while the exit slit was set to 5 nm. Protein concentration was measured before each experiment and MF2 concentration was determined using an extinction coefficient at 369 nm of 22,000 M^−1^ × cm^−1^ [20]. In each experiment, 1.0 μM apoprotein and 0.5 μM MF2 in binding buffer (20 mM HEPES pH 7.4, 200 mM NaCl, 5% *v*/*v* glycerol) were titrated with increasing concentrations of ZnCl_2_, keeping the total volume of titrant added to less than 5% *v/v*. Fluorescence excitation spectra were scanned from 250–450 nm while monitoring emission at 505 nm. Experiments were performed in triplicate and the fluorescence intensities at λ_ex_ = 330 nm were fit using the program DYNAFIT v. 4.05.103 [40,41] using scripts adapted from Golynskiy et al. [20]. Prior to each series of experiments, the affinity of MF2 for zinc in our buffer system was determined using DYNAFIT and used in our calculation of protein binding affinity. From 4 independent measurements, the Kd of MF2 for zinc was 30 +/− 12 nM, consistent with the literature value of 36 nM [20].

### 4.9. Equilibrium Zinc Binding Assay

Apo WT or mutant ZrgA at 100 μM in binding buffer (20 mM HEPES pH 7.4, 200 mM NaCl, 5% *v/v* glycerol) was combined with various concentrations of ZnCl_2_ and incubated for 15 min at room temperature. Samples were then desalted into binding buffer using Zeba™ desalting columns (Thermo Scientific, Waltham, MA, USA) and the final protein concentration was determined by absorbance at 280 nm. Samples were digested in nitric acid, diluted, and zinc concentrations were determined by ICP-OES as described above.

### 4.10. Crystallization and Structure Determination

Initial crystallization hits for apo Δ124-184 ZrgA were identified using the Hauptman–Woodward Institute standard screen [42]. These were optimized in-house, and diffraction-quality crystals were grown under paraffin oil using a 1:1 ratio of 10 mg mL^−1^ apo Δ124-184 ZrgA and precipitant solution composed of 0.1 *M* bis-tris propane pH 5.5, 0.2 *M* sodium chloride, and 28% PEG 3350. The crystals were cryoprotected in a precipitant solution containing 10% PEG 400 and were cryocooled in liquid nitrogen. To obtain the zinc-bound structure, a protein crystal grown in 0.1 *M* sodium acetate pH 4.6, 0.2 *M* ammonium acetate, and 27% PEG 4000 was soaked for approximately 1 min in precipitant solution containing 10% PEG 400 and 1 mM ZnCl_2_ prior to cryocooling.

Diffraction data were collected at 100 K on beamline 5.0.2 at the Advanced Light Source at Berkeley National Laboratory, indexed and integrated with XDS version 0.92 [43,44] and scaled using AIMLESS [45]. Data collection and processing statistics are summarized in Table 2. The model of apo PA4063 (PDB entry 7AHW [17]) lacking waters was used as the search model for the molecular replacement of apo ZrgA using Phaser-MR [46] whereas the apo ZrgA structure was used to solve the zinc-bound structure. Manual model building was performed in Coot [47] and further rounds of refinement were performed in Phenix [48] with final refinement using REFMAC [49]. Atomic coordinates of apo and zinc-bound Δ124-184 ZrgA have been deposited in the PDB as entries 8EZW and 8F1B, respectively. Figures were prepared using Pymol version 2.5.4, Schrödinger, LLC. New York, NY, USA., which was also used for pairwise structural alignments.

## Figures and Tables

**Figure 1 ijms-24-00548-f001:**
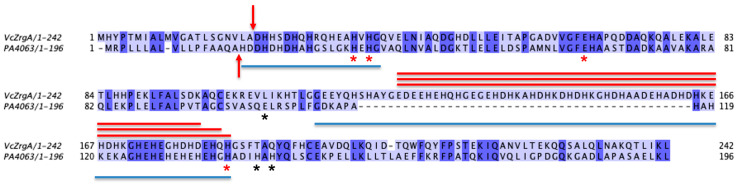
Sequence alignment of ZrgA homologues from *V. cholerae* (VcZrgA) and *P. aeruginosa* (PA4063). Arrows indicate predicted cleavage sites for the N-terminal periplasmic targeting sequence [18,19]. Asterisks indicate zinc binding residues in PA4063 colored by binding site (site a, black; site b, red). Blue lines under the PA4063 sequence indicate unstructured regions in the apo crystal structure. Red lines above the VcZrgA sequence indicate regions deleted in mutants generated in this work.

**Figure 2 ijms-24-00548-f002:**
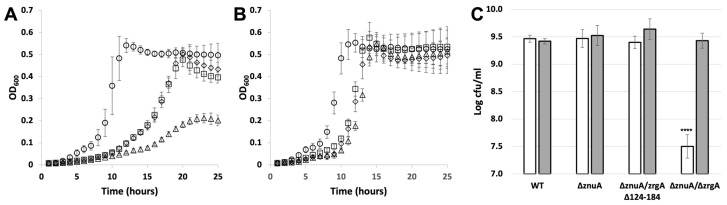
Growth and survival of WT and mutant *V. cholerae*. Growth curves for WT (circles), *ΔznuA* (squares), *ΔznuA*/*zrgA Δ124-184* (diamonds), *ΔznuA/ΔzrgA* (triangles) in the absence of zinc (**A**) or in the presence of 50 μM ZnCl_2_ (**B**). Data are representative of 3 replicate experiments. Error bars represent standard deviations for 9 replicate wells. (**C**) Colony forming units (CFU) for strains grown for 24 h under zinc-free (white bars) and 50 μM ZnCl_2_ (gray bars) conditions. Error bars represent standard deviations for 3 replicate plating experiments. A two-way ANOVA followed by Tukey’s multiple comparison test demonstrates that viability in the *ΔznuA/ΔzrgA* strain under zinc-deficient conditions was significantly decreased relative to all other strains and conditions, **** *p* < 0.0001.

**Figure 3 ijms-24-00548-f003:**
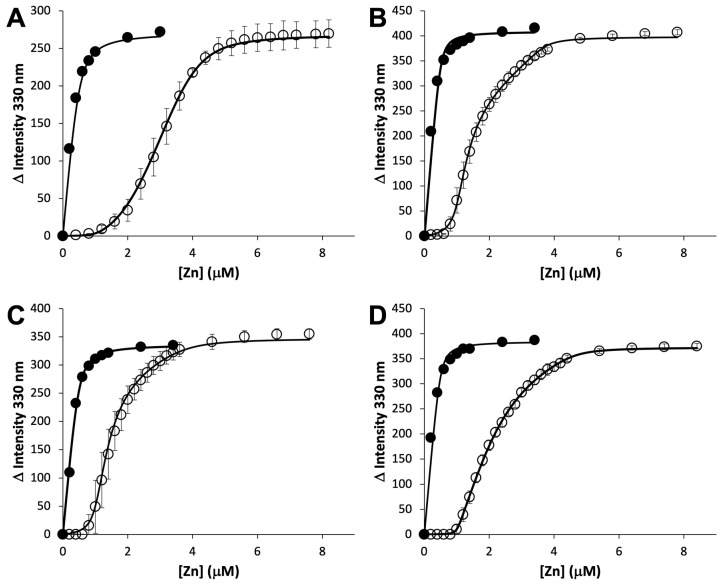
Competitive assay for zinc binding. MagFura-2 at 0.5 μM in the absence (filled circles) or presence (empty circles) of 1.0 μM WT ZrgA (**A**), Δ124-184 ZrgA (**B**), Δ124-183 ZrgA (**C**) or Δ124-180 ZrgA (**D**) was titrated with ZnCl_2_. The change in fluorescence intensity was tracked at 330 nm representing formation of the MF2-Zn complex and plotted against total added zinc. Each experiment was run in triplicate and error bars represent the standard deviation between experiments. Fits with parameters indicated in Table 1 are shown as solid lines.

**Figure 4 ijms-24-00548-f004:**
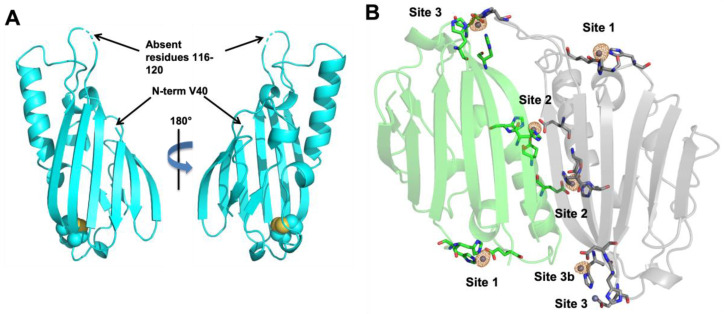
Crystal structures of Δ124-184 ZrgA. (**A**) Apo protein with backbone shown in cartoon format and Cys residues comprising the disulfide bond shown as spheres. (**B**) Protein bound to zinc. Chains A (green) and D (gray) are shown. Backbone atoms are shown as cartoon, zinc ligands are shown as sticks colored according to element and zinc ions are shown as spheres. The anomalous difference density map (orange mesh) is contoured to 5σ showing the positions of zinc ions. Figure created with Pymol version 2.5.4, Schrödinger, LLC. New York, NY, USA.

**Figure 5 ijms-24-00548-f005:**
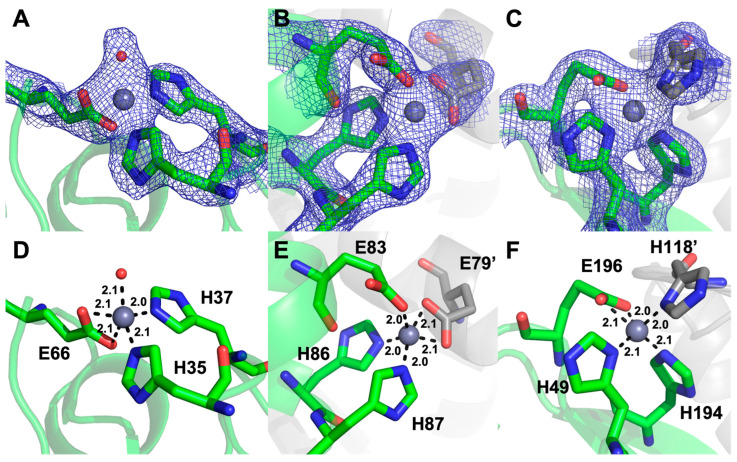
Zinc binding sites in Δ124-184 ZrgA showing 2Fo-Fc density contoured at 1.5σ (**A**–**C**) and zinc ligands with bond distances indicated in Å (**D**–**F**). Site 1 (**A**,**D**), site 2 (**B**,**E**), and site 3 (**C**,**F**) are shown for the A chain (green) with ligands contributed from the D chain (gray) indicated by (’). Figure created with Pymol version 2.5.4, Schrödinger, LLC. New York, NY, USA.

**Figure 6 ijms-24-00548-f006:**
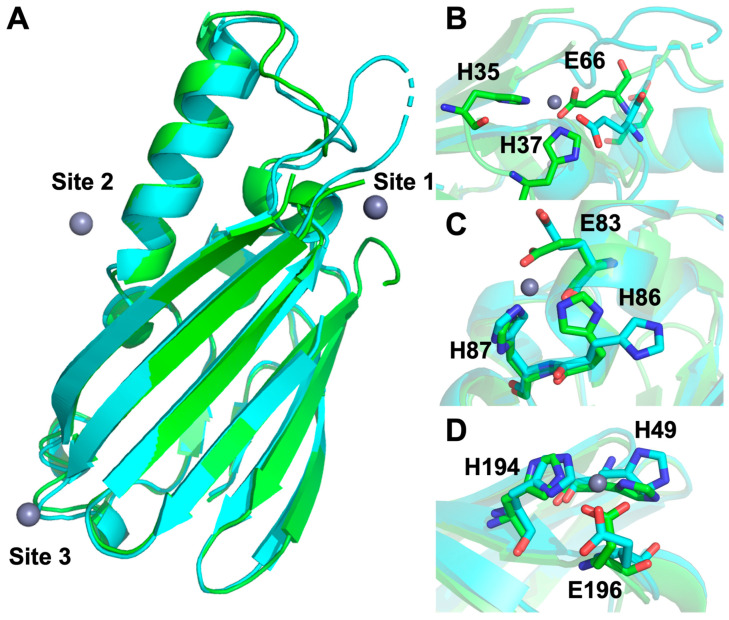
Comparison of zinc-bound (green) and apo (cyan) Δ124-184 ZrgA. The overall structure (**A**) shown as cartoon and details of zinc binding site 1 (**B**), site 2 (**C**), and site 3 (**D**) with zinc ligands shown as sticks colored according to element. Figure created with Pymol version 2.5.4, Schrödinger, LLC. New York, NY, USA.

**Table 1 ijms-24-00548-t001:** Zinc binding affinity and stoichiometry of WT and mutant ZrgA. Uncertainties are expressed as the standard deviations of fitted values for three independent experiments.

Protein	Site	Kd ± S.D. (nM)
WT ZrgA	1	0.1 ± 0.2 *
	2	5.7 ± 4.0
	3	35 ± 22
	4	287 ± 215
Δ124-184 ZrgA	1	0.5 ± 0.3 *
	2	22 ± 3
	3	155 ± 66
Δ124-183 ZrgA	1	0.8 ± 0.7 *
	2	148 ± 126
	3	237 ± 154
Δ124-180 ZrgA	1	0.03 ± 0.06 *
	2	15 ± 4
	3	86 ± 21
	4	111 ± 53

* Values are difficult to estimate as the Kd appears to be near or below the detection limit of this assay (~0.1 nM).

**Table 2 ijms-24-00548-t002:** Data collection and refinement statistics for ZrgA crystal structures.

	Δ124-184 ZrgA Apo	Δ124-184 ZrgA Zinc	Δ124-184 ZrgA Zinc, Zn-Edge
*Data collection*			
Wavelength (Å)	1.000	1.000	1.281
Space group	*P*2_1_	*P*2_1_	*P*2_1_
Unit cell parameters a, b, c (Å) α, β, γ (°)	81.1, 65.4, 88.7 90, 98.8, 90	82.0, 66.9, 88.2 90, 99.9, 90	82.0, 66.9, 88.4 90, 99.8, 90
Resolution range (Å)	45.84–2.00	46.63–2.41	46.65–2.70
Number of reflections (measured/unique)	236,438/61,741	139,225/36,368	99,000/26,001
*R* _merge_	0.06 (0.64)	0.10 (0.69)	0.11 (0.69)
*I/σI*	11.1 (2.2)	9.3 (1.9)	7.1 (1.9)
Completeness (%)	99.2 (98.9)	99.4 (99.8)	99.3 (99.7)
CC_1/2_	1.00 (0.87)	1.00 (0.80)	0.99 (0.83)
Redundancy	3.8 (3.8)	3.8 (3.9)	3.8 (3.9)
*Refinement Statistics*			
Resolution (Å)	45.88–2.00	46.67–2.41	
*R*_work_/*R*_free_	19.2/23.6	21.8/26.2	
*Number of atoms*			
Protein	6381	6666	
Zinc	0	21	
Water	362	96	
Other	20	10	
*R.m.s. deviations*			
Bond lengths (Å)	0.016	0.013	
Bond angles (°)	1.847	1.323	
*Ramachandran Statistics*			
Allowed	99.2%	98.5%	
Outliers	0.8%	1.5%	
Average B-factor (Å^2^)	36.6	40.5	

## Data Availability

The atomic coordinates and structure factors for apo and zinc-bound Δ124-184 ZrgA have been deposited as entry 8EZW and 8F1B, respectively, in the Protein Data Bank, Research Collaboratory for Structural Bioinformatics, Rutgers University, New Brunswick, New Jersey, USA (http://www.rcsb.org/, accessed on 26 December 2022). Other supporting data including spectrum files and growth experiments can be provided upon request to the corresponding author at etyukl@nmsu.edu.

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
