# Peer review of "Structure and Function of the Zinc Binding Protein ZrgA from Vibrio cholerae"

_ijms, 2022, doi:10.3390/ijms24010548_

Round 1

Reviewer 1 Report

The work is devoted to the search for zinc binding sites in ZrgA. 

The main drawback of the work is the inability to link the study of zinc binding to the protein with X-ray data.

The results on the competitive binding of the protein do not allow unambiguous determination of the binding constant, the authors gave data in the range of micromolar concentrations of zinc in the presence of micromolar concentrations of digand and protein, and the evaluation of the constant in the picomolar range is based on the chosen model. A direct method such as ITC can help show such low constants in competitive binding.  Even if we estimate the MF2 binding constant in Figure 3, it appears to be at 200 nM, which is 10 times the known binding constant. 

On the other hand, the authors point out that the protein undergoes significant structural rearrangements induced by zinc binding; it is difficult to understand where the contribution of binding and where the contribution to observed constant  of conformational changes lies. This type of problem can be solved 10.1021/ic500862b . 

X-ray structures were obtained by the authors based on the apo form of the protein followed by saturation of the crystal with zinc ions.  This approach allows the localization of zinc ions in the crystal, but unfortunately does not carry guaranteed information about the natural zinc binding site. 

As a minor remark, I think that coordination of zinc by two oxygen atoms of the carboxy group simultaneously is not possible when we do QM analyzis the electronic structure of the complex; in high-resolution crystals (for example, 6ZPA) it is clearly seen that only one oxygen atom chelates zinc. 

Reviewer 2 Report

This is an interesting research article on the properties of ZrgA from Vibrio cholerae, representative of a newly discovered family of zinc SBPs in ABC transport systems of gram-negative bacteria. It focuses on analysis of growth and survival effects and the zinc-binding properties of V. cholerae ZrgA and a central His-rich sequence region of this homolog relative to the ZrgA from Pseudomonas aeruginosa. The authors show that ZrgA is critical for the zinc-transporting function of the zrgABCDE operon and present evidence on the characterization and assignment of zinc-binding sites in ZrgA using zinc binding assays and crystal structure determination of a ZrgA mutant lacking the central His-rich loop. The paper is well written, novel, well performed and discussed and builds on a series of interesting and contributory papers of the Yukl research team on the structure-function analysis of SBPs linked with zinc ABC transport systems. It is also highly suitable for the IJMS Special Issue on Metal transporters in health and disease. 

I have only some minor comments for improvement.

Line 124; the range is 0.03 – 287 nm (based on Table 1), not 0.05 – 287 nm.

Line 133, the starting concentrations are 200 μM, not 100 μM (based on Fig. S2)

Line 246, based on the deletion mutant assayed (Fig. S2) it should be residues 124-184, not 124-180.

Table 1, column “Protein”, the annotation Delta120-180 ZrgA is wrong. Based on the data provided, it should be Delta124-180 ZrgA (Fig. 1, legend to Fig. 3, line 122, line 248, line 352). However, it is also annotated as Delta120-180 (Table 1) or Delta120-184 (line 114); and in Suppl. Table 2 (fifth row in the list) as Delta123-180.

There is some inconsistency with the naming of the Delta124-183 deletion mutant, as well. It is given as Delta124-183 in most cases (Fig. 1, legend to Fig. 3, line 352), but occasionally it is annotated as Delta123-183 (Table 1) or Delta123-184 (line 114, and Suppl. Table 2).   

Line 151, “the protein is purified exclusively as a monomer as determined by size exclusion chromatography”; these data are not shown in the manuscript; it could be shown as a figure in the Supplement

Lines 121-123, “The data for wild-type and Delta124-180 ZrgA best fit a model with 4 high-affinity binding sites, while only 3 sites were required to fit the data for the other two, longer deletion mutants (Delta124-183, Delta124-184)”. Does this mean that the additional high-affinity zinc binding site involves His182 or Glu181? (Fig. 1).

The authors conclude that the His-rich unstructured region “provides additional zinc binding sites in vitro” (Abstract, lines 22-23). In Discussion, lines 245-258, they state: “In vitro, wild-type ZrgA can bind up to 10 zinc ions (Fig. S2) with four high-affinity (Kd < 1 μM) zinc binding sites (Table 1). The unstructured, His-rich loop residues 124-184 account for four or five binding sites (Fig. S2), one of which is high-affinity as indicated by the preservation of all four high affinity binding sites in the Delta124-180 mutant”. These sentences need to be written more clearly. For example, three, not four, high affinity sites are found with the MF2 assay in the Delta124-184 mutant.   

In the light of the authors’ statement that ZrgA represents a new family of zinc SBPs widespread in gram-negative bacteria (Abstract, lines 16-17), I would suggest that the authors provide some more information or discussion on the potential evolutionary implications of the presence of this additional zinc-regulated ABC operon in several bacteria. The relevant information given in the text is rather sparse (Introduction, lines 55-56; Discussion, lines 300-302; and a reference to the authors’ previous work [ref. 16 in the manuscript] which provides an interesting bioinformatic analysis but not a discussion on the evolutionary implications of the findings).

In References, all species names should be italicized. They are not. 

Line 352, zrgA should be italicized.  

Line 223 (legend to Fig. 6), correct Zrg to ZrgA.
